# Unsupervised Grammatical Error Correction Rivaling Supervised Methods

**Hannan Cao**[1*]  **Liping Yuan**[2]  **Yuchen Zhang**[2]  **Hwee Tou Ng**[1]

[1]Department of Computer Science, National University of Singapore
[2]ByteDance
caoh@u.nus.edu, yuanliping.0o0@bytedance.com
zhangyuchen.zyc@bytedance.com, nght@comp.nus.edu.sg

## Abstract

State-of-the-art grammatical error correction (GEC) systems rely on parallel training data (ungrammatical sentences and their manually corrected counterparts), which are expensive to construct. In this paper, we employ the Break-It-Fix-It (BIFI) method to build an unsupervised GEC system. The BIFI framework generates parallel data from unlabeled text using a fixer to transform ungrammatical sentences into grammatical ones, and a critic to predict sentence grammaticality. We present an unsupervised approach to build the fixer and the critic, and an algorithm that allows them to iteratively improve each other. We evaluate our unsupervised GEC system on English and Chinese GEC. Empirical results show that our GEC system outperforms previous unsupervised GEC systems, and achieves performance comparable to supervised GEC systems without ensemble. Furthermore, when combined with labeled training data, our system achieves new state-of-the-art results on the CoNLL-2014 and NLPCC-2018 test sets.[1]

## 1 Introduction

Grammatical Error Correction (GEC) (Chollampatt et al., 2016; Chollampatt and Ng, 2018; Qorib et al., 2022; Bryant et al., 2023) is the task of correcting errors in a source sentence and generating a grammatically correct target sentence. Current state-of-the-art (SOTA) systems (Rothe et al., 2021) have reached good performance using sequence-to-sequence (seq2seq) models. However, a common drawback of these systems is their extensive reliance on a significant quantity of labeled data. For instance, Rothe et al. (2021) utilized over 2 million sentence pairs, which are time-consuming and costly to obtain as they require human manual correction. Unsupervised GEC systems aim to over-

come this limitation. However, the current performance of unsupervised GEC systems (Alikaniotis and Raheja, 2019; Yasunaga et al., 2021) is much lower than supervised systems. Moreover, they still require manually defined or extracted confusion sets to generate synthetic data and assess sentence grammaticality. As a result, this greatly hinders the applicability of unsupervised GEC systems.

The SOTA unsupervised GEC system, LM-critic (Yasunaga et al., 2021), uses the Break-It-Fix-It (BIFI) framework (Yasunaga and Liang, 2021) to extract realistic parallel data from unlabeled data. Specifically, the BIFI framework utilizes a *fixer* and a *critic*. The fixer is designed to perform the GEC task, while the critic is designed for the grammatical error detection (GED) task, which classifies an input sentence as grammatical or ungrammatical. Given a critic which classifies each unlabeled sentence as grammatical or ungrammatical, BIFI generates parallel data to train a better fixer by the following four steps. (1) Correct ungrammatical sentences with the existing fixer and collect outputs that are classified as grammatical by the critic. (2) Train a grammatical error generator (called a *breaker*) using the sentence pairs obtained in (1). (3) Corrupt the grammatical sentences with the breaker and collect the outputs that the critic classifies as ungrammatical. (4) Obtain parallel data by combining outputs of (1) and (3). LM-Critic uses local neighborhood information and perplexity (PPL) to build the critic and uses synthetic data to initialize the fixer. However, the synthetic data relies on the edit pairs provided by Awasthi et al. (2019), which are extracted from labeled sentences. Moreover, a significant performance gap remains between LM-critic and supervised systems (See Section 4).

In this paper, we propose a novel method for generating synthetic data and building a critic, with the aim of building an unsupervised GEC system that can rival supervised systems. By examining

---

*Work done during Cao's internship at ByteDance.

[1]Source code available at https://github.com/nusnlp/ugec.

the grammatical errors in labeled data, we identified several language-independent error patterns. Using these patterns, we propose a synthetic data generation method based on a masked language model (MLM) to build a fixer. Subsequently, we use this fixer as a basis for building our critic. The critic is trained using grammaticality labels obtained from high-confidence fixer predictions. To address the data scarcity problem that arises from high-confidence filtering, we propose a masking-based approach and a self-knowledge distillation method for data augmentation. The unsupervised GEC system is trained using the BIFI framework, with the fixer and the critic being refined repeatedly in iterations.

We evaluate the performance of our system on both English and Chinese GEC tasks. Specifically, we evaluate our system on the CoNLL-2014 (Ng et al., 2014) and BEA-2019 (Bryant et al., 2019) test sets for English GEC, and on the NLPCC-2018 (Zhao et al., 2018) test set for Chinese GEC. Our unsupervised system outperforms the prior unsupervised SOTA by 12.5 $F_{0.5}$ and 13.8 $F_{0.5}$ on the CoNLL-2014 and BEA-2019 test sets, respectively. Our unsupervised system also compares favorably with the best-performing supervised systems for both languages. Furthermore, when we further train our system with labeled data, we surpass the SOTA results on both CoNLL-2014 and NLPCC-2018 test sets.

The contributions of our paper are as follows:

- We introduce a novel method for unsupervised synthetic data generation, based on MLM and language-independent error patterns. Compared to existing approaches, our method generates more realistic synthetic data, and provides a better unsupervised fixer.

- We propose a new method to build an unsupervised critic with high-confidence predictions from the fixer model. This approach enables the critic model to continually enhance its performance over iterations, demonstrating better performance than prior methods.

## 2   Related Work

**Unsupervised grammatical error correction.** Prior research (Alikaniotis and Raheja, 2019) builds an unsupervised GEC system by leveraging manually constructed confusion sets to provide possible corrections, and uses language models (LMs)

to validate these corrections. Yasunaga et al. (2021) utilize the confusion sets and LM in a different way. Instead of constructing a GEC model directly, Yasunaga et al. (2021) use them to create a GED model. This GED model is then combined with the BIFI method to build an unsupervised GEC system. In contrast to these works, our method does not rely on any manually constructed confusion sets, making it easy to extend to low-resource languages.

**Synthetic data generation.** Synthetic data generation for GEC commonly adopts two strategies: backtranslation-based corruption methods using labeled data (Kiyono et al., 2019; Stahlberg and Kumar, 2021; Xie et al., 2018), and error injection corruption methods via edit pairs or confusion sets extracted from labeled data (Awasthi et al., 2019; Lichtarge et al., 2019; Yuan and Felice, 2013). Methods that do not require labeled GEC data have been explored by Grundkiewicz et al. (2019) and Sun et al. (2022). The former utilizes spellchecker-based confusion sets to generate erroneous sentences, while the latter applies machine translation pairs and a pre-trained cross-lingual language model (XLM) for sentence corruption. Our method avoids external dependencies, such as confusion sets, spellcheckers, or translation pairs.

**Text evaluation.** Prior work in GEC (Bryant et al., 2019; Dahlmeier and Ng, 2012; Niu and Penn, 2020) assesses sentence grammaticality through reference text or syntactic information, such as part-of-speech tags. Yasunaga et al. (2021) mitigate this reliance with an LM-based method, yet it still needs pre-defined confusion sets. Our method constructs a critic using high-confidence predictions from the fixer model, thereby completely eliminating the need for external information.

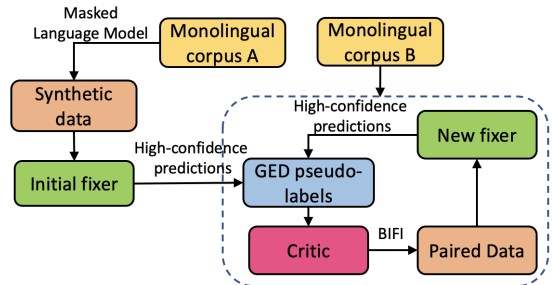

Figure 1: Our unsupervised GEC system involves the following four steps. (1) Create an initial fixer with the synthetic data generated through an MLM-based method. (2) Construct a critic based on high-confidence predictions from the fixer. (3) Build a new fixer using the parallel data extracted by BIFI. (4) Repeat steps 2 and 3 until the fixer's performance converges.

## 3 Method

Figure 1 illustrates our method to build an unsupervised GEC system. It contains two key components: initial fixer[2] construction (§3.2) and the critic construction (§3.3).

### 3.1 Problem Setup

Grammatical error correction aims to correct an ungrammatical sentence $x^{(i)}$ into its grammatical version $y^{(i)}$ while preserving the original semantics. In the supervised setting with annotated data available, the GEC model leverages labeled sentence pairs $D_l = \{(x^{(i)}, y^{(i)})\}$ to learn a mapping from $x$ to $y$. However, in the unsupervised setting, the GEC model must infer this mapping from a monolingual corpus $D_m = \{x^{(i)}\}$. The BIFI framework offers a mechanism to extract realistic parallel data from unlabeled sentences using a fixer $f$ and a critic $c$. The fixer maps $x$ to $y$, and the critic evaluates the grammaticality of a given sentence. Our goal is to construct a good initial fixer $f_0$ (§3.2) and critic (§3.3) through unsupervised methods and utilize them to develop the final fixer $f_n$ (§3.4).

### 3.2 Training an Initial Fixer

The BIFI framework relies on a good initial fixer $f_0$. Intuitively, $f_0$ could be obtained by training a model with synthetic data generated via unsupervised approaches. However, how to generate realistic synthetic data without reliance on supervised information (e.g., edit pairs) remains an open problem. To tackle this problem, we analyze the parallel data in English and Chinese to identify some language-independent error patterns (§3.2.1). Leveraging these patterns, we propose an unsupervised synthetic data generation method (§3.2.2).

### 3.2.1 Exploiting Error Patterns

We carry out analysis on the GEC validation set and categorize the errors into three categories: insertion errors, deletion errors, and replacement errors. Inspired by context-free spell-checkers, we plot the edit distance distribution between erroneous source tokens and their corresponding target tokens for replacement errors. For both deletion and insertion errors, we plot the frequency distribution of each erroneous token of the vocabulary.

As depicted in Figure 2, it is evident that the edit distance between an erroneous token and its

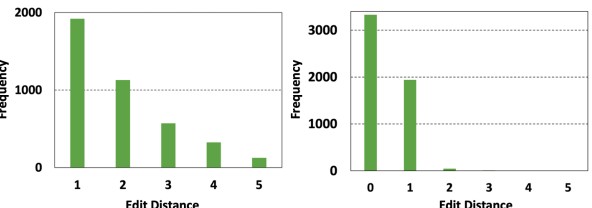

Figure 2: The character-level edit distance between an erroneous token and its corresponding target token for replacement errors. *Left:* For English, we compute the character-level edit distance directly. *Right:* For Chinese, we convert the tokens into Pinyin before computing the character-level edit distance. Instances where the edit distance equals 0 are due to homophonic tokens.

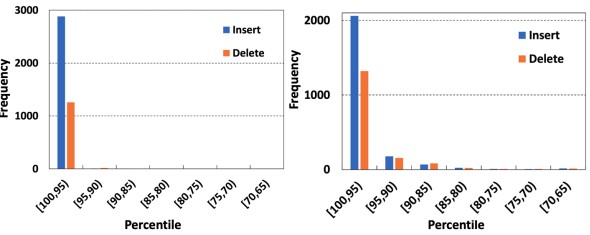

Figure 3: The erroneous token distribution for insertion and deletion errors. The tokens of the vocabulary are ordered by decreasing frequency. *Left:* English; *Right:* Chinese.

target token is typically small for both English and Chinese replacement errors. In either language, the majority of the edit distances are confined by the typical length of a "word". In Figure 3, we can see that the vast majority of incorrect tokens resulting from insertion and deletion errors are found within the top 5% of the vocabulary. This leads to the conclusion that these errors are commonly associated with high-frequency tokens. Based on these observations, we define two language-independent error patterns:

**Replacement errors.** The edit distance between an erroneous token and its corresponding target token is typically small.

**Insertion and deletion errors.** The erroneous token usually has a high frequency in the vocabulary.

Leveraging these two patterns, we outline our unsupervised synthetic data generation approach in §3.2.2.

### 3.2.2 Unsupervised Synthetic Data Generation

We synthesize erroneous sentences from a clean corpus using the following steps: for each sentence $x^{(i)}$ from the seed corpus $D_m^{seed}$, we first sample the error count per sentence from a pre-defined dis-

---

[2]To differentiate between the fixer obtained via synthetic data and the fixer obtained via paired data through BIFI, we name the former fixer as the initial fixer.

tribution (Awasthi et al., 2019). We introduce each error by performing one of these three operations: (1) delete a token $w_v \in x^{(i)}$ with probability $p_{del}$; (2) insert a token $w_v$ at a random position with probability $p_{ins}$; (3) replace a token $w_j \in x^{(i)}$ with $w_r$ by probability $p_{rep}$.[3]

Specifically, to generate the replacement token $w_r$, we replace a randomly selected token $w_j \in x^{(i)}$ with the mask token [MASK] and utilize MLM to predict a set of candidate tokens at the masked position based on its surrounding context. In this work, we choose RoBERTa as the MLM in our implementation. As described in Section 3.2.1, only candidates with a low edit distance from $w_j$ are appropriate replacements. Therefore, we eliminate candidate tokens that have an edit distance exceeding a certain threshold. Finally, we sample $w_r$ from the remaining candidates using a pre-defined distribution solely based on the edit distance.

To circumvent the problem of consistently sampling the same high-frequency tokens for insertion and deletion errors, we design a smoothing function to smooth the frequency of tokens in the vocabulary. This process is detailed in Algorithm 1. In Algorithm 1, $LIST_{ID}$ represents a list of breakpoints $(id_i)$, which are positive integers in ascending order used for comparing against the rank of a token. Note that the tokens of the vocabulary are organized in descending order of frequency, where a token with a smaller rank occurs more frequently. This design ensures that high-frequency tokens in a collection possess an equal chance of being sampled, while maintaining a higher frequency than the less frequent tokens. We diverge from sampling based on the raw frequency of tokens in the vocabulary, opting to sample according to the smoothed frequency $f_{smooth}$.

---

**Algorithm 1:** Smoothing Function

**Input:** $LIST_{ID} = [id_0, id_1, ... , id_n]$
      $w_v$: a token in the vocabulary
**Output:** Smoothed probability $f_{smooth}$ of $w_v$
1: Find the rank $k$ for $w_v$ in the vocabulary
2: Find the smallest $i$ such that $k \leq id_i$
3: **if** $i = 0$ **then**
4:     $f_{smooth} = 1/id_0$
5: **else**
6:     $f_{smooth} = 1/(id_i - id_{i-1})$
7: **end if**

---

[3]The sum of $p_{del}$, $p_{ins}$, and $p_{rep}$ equals to one

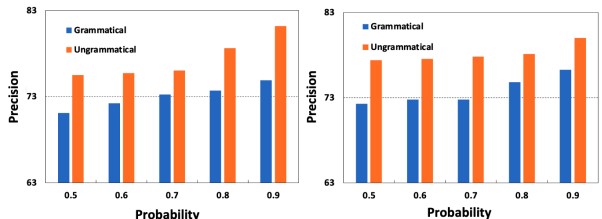

Figure 4: Correlation between the probability of producing $\hat{y}^{(i)}$ and precision of $z^{(i)}$. *Left:* English; *Right:* Chinese.

### 3.3 Training a Critic

LM-Critic integrates word-level perturbations with sentence perplexity to define the critic. However, the efficacy of word-level perturbations relies on pre-defined confusion sets. To circumvent this reliance, an intuitive approach is to extract the GED pseudo-labels from the existing fixer and then train a binary classifier from such pseudo-labels as the critic. Specifically, we begin by randomly choosing a subset $D'_m$ from $D_m$. For each sentence $x^{(i)} \in D'_m$, we use the fixer to make corrections and obtain the output $\hat{y}^{(i)}$. If $\hat{y}^{(i)}$ is different from $x^{(i)}$, then we assign a pseudo-label $z^{(i)} = 0$, meaning that $x^{(i)}$ is "ungrammatical". Otherwise, we assign $z^{(i)} = 1$, meaning that $x^{(i)}$ is "grammatical".

Since the initial fixer is far from optimal, the pseudo-labels assigned by the initial fixer may have low precision. To address this problem, we analyze the relation between the confidence of $\hat{y}^{(i)}$ and the precision of $z^{(i)}$. In Figure 4, we observe that high-confidence predictions (i.e., $\hat{y}^{(i)}$ predicted with a high probability) are associated with more accurate grammaticality labels. Therefore, we propose to select a highly confident subset $D_{sub}$ from $D'_m$ such that for every $x^{(i)} \in D_{sub}$, the fixer predicts $\hat{y}^{(i)}$ with probability greater than 0.9.

It is worth noting that when the critic is trained on fixer predictions, it may unintentionally cause over-fitting to the fixer, which undermines the critic's ability to enhance the fixer further through iterations. Xie et al. (2020) has demonstrated the importance of introducing noise throughout the self-training process. Accordingly, we propose a masking-based data augmentation approach when building the critic. Specifically, for each sentence $x^{(i)} \in D_{sub}$, we generate an augmented sentence $x^{(i)}_{\text{masked}}$ by randomly replacing $p\%$ tokens with the [MASK] token, and minimize the loss function $L_{\text{masked}}$ with respect to the critic's model parame-

ters $\theta_{cr}$:

$$L_{\text{masked}} = -\frac{1}{|D_{sub}|} \sum_{x^{(i)} \in D_{sub}} \sum_{c \in \{0,1\}} \mathbb{1}\{z^{(i)} = c\} \cdot$$
$$(\log P(c|x^{(i)}; \theta_{cr}) + \log P(c|x^{(i)}_{\text{masked}}; \theta_{cr})) \quad (1)$$

Another issue of selecting high-confidence pseudo-labels is data scarcity. With the initial fixer, only 20% of the sentences from $D'_m$ are selected. To mitigate this issue, we utilize a self-knowledge distillation (SKD) technique to gather additional training data and enhance the model's generalizability. Specifically, for each $x^{(i)} \in D'_m$, we follow the method used by (Xie et al., 2016; Meng et al., 2020) to construct soft pseudo-labels $\tilde{z}_c^{(i)4}$:

$$\tilde{z}_c^{(i)} = \frac{[P(c|x^{(i)}; \theta'_{cr})]^2/f_c}{\sum_{c' \in \{0,1\}}\{[(P(c'|x^{(i)}; \theta'_{cr})]^2/f_{c'}\}} \quad (2)$$

where $f_c = \sum_{x^{(i)} \in D_{m'}} P(c|x^{(i)}; \theta'_{cr})$ is the sum over soft frequencies for class $c$, and $\theta'_{cr}$ is the critic's model parameters in the previous epoch. In the first epoch, $\theta'_{cr}$ represents the critic's model parameters obtained by minimizing (1). Once the soft pseudo-labels are obtained, we train a new critic model by minimizing the following loss function:

$$L_{\text{critic}} = L_{\text{masked}} + L_{\text{skd}} \quad \text{where}$$
$$L_{\text{skd}} = -\frac{1}{|D_{m'}|} \sum_{x^{(i)} \in D_{m'}} \sum_{c \in \{0,1\}} \tilde{z}_c^{(i)} \cdot \quad (3)$$
$$(\log P(c|x^{(i)}; \theta_{cr}) + \log P(c|x^{(i)}_{\text{masked}}; \theta_{cr}))$$

---

**Algorithm 2:** Break-It-Fix-It (BIFI)

**Input:** Fixer $f$, critic $c$, grammatical sentences $D_m^g$, and ungrammatical sentences $D_m^{ug}$

**Output:** (erroneous, corrected) sentence pairs.
1: Correct $D_m^{ug}$ using the fixer $f$ and retain output deemed grammatical by the critic $c$.
2: Train a breaker (error generator) on the resulting paired data.
3: Corrupt $D_m^g$ using the breaker and retain output deemed ungrammatical by the critic $c$.
4: Combine the parallel data obtained in Step 1 and 3.

---

[4]The intuition is to (1) strengthen predictions; (2) emphasize data with high confidence; and (3) normalize the loss contribution of each centroid. Refer to (Xie et al., 2016) for details.

## 3.4 Iteratively Refining the Fixer and Critic

Algorithm 3 provides a high-level overview of our unsupervised grammatical error correction (GEC) system. We start by applying the unsupervised technique outlined in §3.2.2 to corrupt $D_m^{seed}$ and yield synthetic data. This synthetic data is then employed to train an initial fixer, denoted by $f_0$. In the next phase, we leverage $f_0$ and $D_m$ to derive pseudo labels and train a RoBERTa-based critic, as described in §3.3. By utilizing this critic, we segregate $D_m$ into grammatically correct ($D_m^g$) and incorrect ($D_m^{ug}$) subsets. We then use the BIFI mechanism to generate realistic parallel data that is then employed to train a new fixer $f_1$. We subsequently substitute $f_0$ with $f_1$ and repeat this procedure until the fixer achieves satisfactory performance.

---

**Algorithm 3:** Unsupervised GEC system

**Input:** Monolingual corpora $D_m^{seed}$, $D_m$
1: Generate synthetic data using the method described in §3.2.2 to corrupt $D_m^{seed}$
2: Build $f_0$ with synthetic data
3: **for** $t = 1, 2, 3, \ldots$ **do**
4:     Extract GED pseudo-labels with $f_{t-1}$
5:     Train a critic (grammaticality classifier) by minimizing Eqn (3), then use it to split $D_m$ into $D_m^g$ and $D_m^{ug}$
6:     Use BIFI (Algorithm 2) to generate parallel data to train a new fixer $f_t$.
7: **end for**

---

## 4 Experiments on English GEC

### 4.1 Data and Model Configuration

Following prior work (Awasthi et al., 2019; Grundkiewicz et al., 2019), we use the combination of WMT NewsCrawl corpus (Bojar et al., 2018) and One-Billion-Word corpus (Chelba et al., 2014) as the seed monolingual corpus $D_m^{seed}$. We generate 145 million synthetic sentence pairs with the method described in §3.2.2. These synthetic pairs are used to fine-tune the Flan-T5-xxl model (Chung et al., 2022) to create the initial fixer $f_0$.

Following Yasunaga et al. (2021), our monolingual dataset $D_m$ contains both grammatical and ungrammatical sentences. Concretely, we randomly select 10 million unlabeled sentences from various sources: Yahoo!Answer corpus (Zhang et al., 2015), Wikipedia history (Grundkiewicz and Junczys-Dowmunt, 2014), Lang8 (Mizumoto et al.,

| System | Architecture | Ens | CoNLL-2014 | | | BEA-2019 | | |
|---|---|---|---|---|---|---|---|---|
| | | | P | R | $F_{0.5}$ | P | R | $F_{0.5}$ |
| *Unsupervised System* | | | | | | | | |
| Alikaniotis and Raheja (2019) | GPT2 | ✗ | 58.5 | 24.9 | 46.1 | - | - | - |
| Grundkiewicz et al. (2019)* | BART-base | ✗ | 59.7 | 18.5 | 41.3 | 62.4 | 25.4 | 48.8 |
| Yasunaga et al. (2021)* | BART-base | ✗ | 64.4 | 35.6 | 55.5 | 67.3 | 46.1 | 61.6 |
| ChatGPT with prompting† | ChatGPT | ✗ | 50.3 | *59.7* | 51.9 | 42.6 | 69.3 | 46.1 |
| GPT4 with prompting† | GPT4 | ✗ | 60.8 | 57.0 | 59.9 | 56.8 | *70.2* | 59.1 |
| *Supervised System* | | | | | | | | |
| Sorokin (2022) | RoBERTa-large | ✗ | 79.4 | 36.1 | 64.0 | 86.2 | 54.2 | 77.1 |
| Zhang et al. (2022b) | BART-large | ✗ | 74.7 | 49.0 | 67.6 | 75.1 | 65.5 | 72.9 |
| Rothe et al. (2021) | T5-xxl | ✗ | - | - | 68.9 | - | - | 75.9 |
| Lai et al. (2022) | - | ✓ | 78.2 | 42.7 | 67.0 | - | - | 77.9 |
| Qorib et al. (2022) | - | ✓ | **81.5** | 43.8 | 69.5 | **86.7** | 60.9 | **79.9** |
| *Our Unsupervised System* | | | | | | | | |
| Initial fixer | BART-base | ✗ | 66.2 | 35.8 | 56.6 | 63.1 | 41.3 | 57.1 |
| 1st iteration | BART-base | ✗ | 67.2 | 40.2 | 59.3 | 68.3 | 48.8 | 63.2 |
| 2nd iteration | BART-base | ✗ | 69.3 | 40.5 | 60.6 | 67.2 | 51.7 | 63.4 |
| 3rd iteration | BART-base | ✗ | 66.8 | 44.5 | 60.7 | 65.6 | 57.4 | 63.8 |
| Initial fixer | Flan-T5-xxl | ✗ | 70.0 | 36.5 | 59.1 | 73.1 | 52.1 | 67.6 |
| + supervised data | Flan-T5-xxl | ✗ | 74.5 | 53.6 | 69.1 | 78.6 | 67.8 | 76.1 |
| 1st iteration | Flan-T5-xxl | ✗ | 75.5 | 42.0 | 65.2 | 79.5 | 55.4 | 73.1 |
| 2nd iteration | Flan-T5-xxl | ✗ | 75.6 | 45.6 | 66.8 | 80.5 | 57.8 | 74.6 |
| 3rd iteration | Flan-T5-xxl | ✗ | *74.9* | 49.6 | *68.0* | *79.6* | 62.5 | *75.4* |
| + supervised data | Flan-T5-xxl | ✗ | 75.0 | **53.8** | **69.6#** | 78.8 | **68.5** | 76.5 |

Table 1: Performance (in %) of GEC systems on English GEC test sets. Ens: indicates if the system uses the ensemble method. *: represents our reproduced result. Specifically, Grundkiewicz et al. (2019) proposed an unsupervised synthetic data generation method. We use this synthetic data to train the BART-base model to make a fair comparison to LM-Critic and our unsupervised system. †: The zero-shot performance of ChatGPT and GPT4 using the best prompt from (Coyne et al., 2023). The best unsupervised and supervised results are shown in ***bold*** and **bold**, respectively. Statistically significant improvements ($p < 0.01$) over the initial fixer + supervised data is marked #.

2011), NUCLE (Dahlmeier et al., 2013), and FCE (Yannakoudakis et al., 2011) datasets. Notably, as Wikipedia history, Lang8, NUCLE, and FCE are labeled datasets, we only take sentences from the source side of these datasets[5]. When constructing the critic, we use the Lang8 dataset as $D'_m$ and choose RoBERTa-base as our classifier model.

We evaluate the performance of the English GEC system on the CoNLL-2014 and BEA-2019 test sets with the MaxMatch scorer (Dahlmeier and Ng, 2012) and the ERRANT scorer (Bryant et al., 2019), respectively. Following Cao et al. (2021), we use a one-tailed sign test with bootstrap resampling to carry out statistical significance tests. Refer to Appendix A.3 for the detailed experimental settings.

### 4.2 Main Results

Table 1 shows the performance of our system on both CoNLL-2014 and BEA-2019 test sets, including a comparison with existing supervised and unsupervised systems on the leaderboard. Our un-

supervised system achieves $F_{0.5}$ score of 68.0 and 75.4 on the CoNLL-2014 and BEA-2019 test set, respectively, surpassing the current leading unsupervised system (Yasunaga et al., 2021) by 12.5 points on the CoNLL-2014 and 13.8 points on the BEA-2019 test set. Our system also exceeds the zero-shot performance of the GPT4 model by 8.1 points and 16.3 points on the CoNLL-2014 and BEA-2019 test set, respectively. Notably, our system compares favorably with the state-of-the-art supervised single system (Rothe et al., 2021), lagging behind by just 0.9 points on the CoNLL-2014 test set and 0.5 points on the BEA-2019 test set.

To enable a fair comparison with Yasunaga et al. (2021), we replace the Flan-T5-xxl model with the smaller BART-base (Lewis et al., 2020) model when building the fixer. With BART-base, our unsupervised system still outperforms Yasunaga et al. (2021), with a 5.2 $F_{0.5}$ increase on CoNLL-2014 and a 2.2 $F_{0.5}$ increase on BEA-2019. This highlights the superiority of our unsupervised training algorithm.

When we further fine-tune our model using supervised data, the cLang8 (Rothe et al., 2021)

---

[5]The source side sentences are not annotated sentences, and they could be grammatical or ungrammatical.

dataset, our system achieves an $F_{0.5}$ of 69.6 on CoNLL-2014 and 76.5 on BEA-2019. This sets a new SOTA result on the CoNLL-2014 test set.

### 4.3 Analysis

**Synthetic data.** We compare our synthetic data generation method with relevant methods proposed by (Grundkiewicz et al., 2019; Sun et al., 2022), and the method by Awasthi et al. (2019) which was used by (Yasunaga et al., 2021). To enable a fair comparison with the aforementioned data synthesis methods, we randomly select 8 million sentences from the UN Parallel Corpus v1.0 (Ziemski et al., 2016) and corrupt the same monolingual data using each method. We then train a Transformer-base model (Vaswani et al., 2017) on the resulting synthetic data.

| Initial fixer | P | R | $F_{0.5}$ |
|---|---|---|---|
| Spellchecker (Grundkiewicz et al., 2019) | 28.7 | 7.4 | 18.2 |
| Translation pair (Sun et al., 2022) | 31.5 | 8.1 | 19.8 |
| Edit pair (Awasthi et al., 2019) | 39.9 | 11.0 | 25.9 |
| Our method | 38.1 | 12.3 | 26.8 # |
| w/o edit distance | 14.2 | 4.5 | 9.9 |
| w/o high-frequency tokens | 10.2 | 3.9 | 7.7 |

Table 2: Performance of the fixer on the BEA-2019 dev set (Bryant et al., 2019). Statistically significant improvements ($p < 0.01$) over Awasthi et al. (2019) is marked #.

Table 2 shows that our method outperforms competing approaches. As demonstrated in Table 3, the erroneous sentences generated by the competing methods tend to either be grammatically correct or change the intended meaning of the original sentences. This observation explains the better performance of our method relative to these competing approaches. Notably, Sun et al. (2022) implements an approach similar to ours, which also generates replacement errors by inserting masks and then uses XLM to predict the mask. The difference is that they use translation pairs to guide the creation of candidate tokens, while our method relies on edit distance and frequency information.

In our ablation study (Table 2), we find that edit distance and frequency controls are crucial to generate realistic synthetic data, confirming the effectiveness of the error patterns reported in §3.2.1.

**Critic's training methods.** Following (Yasunaga et al., 2021), we randomly sample 600 grammatical sentences and 600 ungrammatical sentences from GEC validation sets and use the averaged $F_{0.5}$ score over 5 runs to measure the performance of the critic. Specifically, to measure the performance across various domains, we assemble our

| Monolingual sentence | Tim mentioned his goal is to discover the hidden spy among us. |
|---|---|
| Grundkiewicz et al. (2019) | Tim mentioned ~~his~~ her goal is to discover the hidden spy among us. |
| Sun et al. (2022) | Tim mentioned his goal is to ~~discover~~ find the hidden spy among us. |
| Awasthi et al. (2019) | Tim mentioned his goal is to discover the hidden spy ~~among~~ between us. |
| Our method | Tim mentioned his goal is ~~to~~ discover the hidden spy among us. |
| Monolingual sentence | During the Second World War the islands were occupied by Germany, causing considerable suffering to the locals. |
| Grundkiewicz et al. (2019) | During the Second World War the islands ~~were~~ was occupied by Germany, causing ~~considerable~~ suffering to the locals . |
| Sun et al. (2022) | During the Second World War the ~~islands~~ isles were occupied by Germany, causing considerable suffering to the locals. |
| Awasthi et al. (2019) | During the Second World War the islands were occupied ~~by~~ Germany, causing considerable suffering time to the locals. |
| Our method | During the Second World War the islands ~~were occupied~~ occupy by Germany, causing considerable suffering to locals. |

Table 3: Example erroneous sentences produced by different approaches.

GEC validation set from the BEA-2019 dev set, the CoNLL-2013 dev set (Ng et al., 2013), and the GMEG-wiki/Yahoo/FCE validation set (Napoles et al., 2019).

We analyze the performance of our critic and compare it to LM-Critic in Table 4. We conduct an ablation study using the following configurations: (1) without employing the self-knowledge distillation method (SKD); (2) without applying the data augmentation approach (DA); and (3) without utilizing the high-confidence subset $D_{sub}$ (CF). Results indicate that all three methods are crucial in enhancing the critic's performance. Notably, our critic outperforms LM-Critic by a significant margin, exhibiting a 13.4 $F_{0.5}$ increase in grammatical and a 14.1 $F_{0.5}$ increase in ungrammatical sentences. Our statistical significance test shows that our critic significantly improves over LM-Critic, and our critic without each individual component (SKD, DA and CF) still significantly improves over LM-Critic.

| Critic | Grammatical | | | Ungrammatical | | |
|---|---|---|---|---|---|---|
| | P | R | $F_{0.5}$ | P | R | $F_{0.5}$ |
| LM-Critic | 63.2 | 76.0 | 65.4 | 69.9 | 55.7 | 66.5 |
| Our Critic | **77.8** | **83.0** | **78.8** | **81.8** | **76.3** | **80.6** |
| w/o SKD | 72.3 | 80.5 | 73.8 | 78.0 | 69.1 | 76.2 |
| w/o DA | 71.0 | 81.7 | 72.9 | 78.5 | 66.6 | 75.7 |
| w/o CF | 68.9 | 81.0 | 71.1 | 76.9 | 64.0 | 73.9 |

Table 4: Performance of our critic (in %) after the 3rd iteration. The ablation study confirms the effectiveness of self-knowledge distillation (SKD), data augmentation (DA) and using high-confidence pseudo labels (CF).

**Fixer's performance through iterations.** In Figure 5, the performance of the fixer across BIFI

iterations is shown. It is observed that the fixer's improvement is stagnant in the absence of the high-confidence subset (CF). Additionally, the fixer's improvement is considerably smaller when data augmentation (DA) or self-knowledge distillation (SKD) is excluded. Moreover, similar to LM-critic, the fixer's improvement comes to a halt after the first iteration without updating the critic. This demonstrates the significance of updating both the critic and the fixer throughout the process.

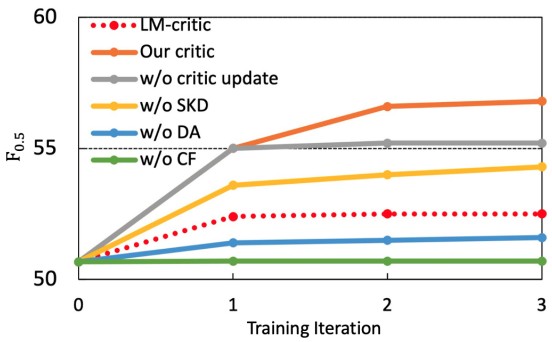

Figure 5: The performance of the fixer across iterations on the BEA-2019 dev set.

**Critic's performance through iterations.** In Figure 6, we observe a consistent improvement in the performance of the critic throughout the iterations. This indicates a mutually beneficial learning process between the critic and the fixer: the critic improves the fixer, which in turn refines the critic even further. The plot on the right shows a correlation between pseudo-label precision and fixer iteration. This suggests that the fixer enhances the critic by providing more accurate GED pseudo-labels.

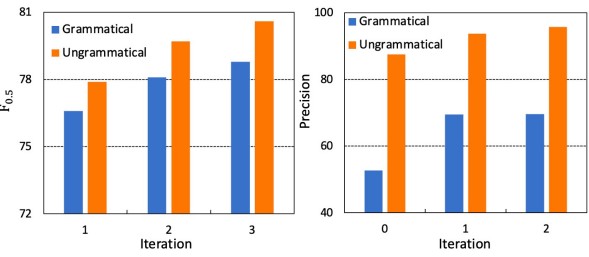

Figure 6: Left: The performance of the critic in different iterations on the BEA-2019 dev set. Right: The precision of $z^{(i)}$ using the fixer in different iterations on the BEA-2019 dev set. Specifically, iteration 0 represents the initial fixer.

**Examples.** In Table 5, we provide qualitative examples to compare the sentences generated by our system with those of GPT4 and LM-Critic. We find that both GPT4 and LM-Critic tend to make unnecessary edits, while our system does

| Input | Keep the information as secret to their spouce is good. |
|---|---|
| GPT4 | Keep the information ~~as~~ a secret ~~to~~ from their spouse is good. |
| LM-Critic | Keep the information as a secret to their spouse is good. |
| Our method | Keep the information secret ~~to~~ from their spouse is good. |
| Input | Laws push a carrier to tell his/her relatives about his problem. |
| GPT4 | Laws push a carrier to tell his/her relatives about ~~his~~ their problem. |
| LM-Critic | Laws push a carrier to tell his/her relatives about ~~his~~ their problem. |
| Our method | Laws push a carrier to tell his/her relatives about his problem . |
| Input | The knowledge of the genetic risk was to be shared within the family. |
| GPT4 | The knowledge of the genetic risk ~~was~~ were to be shared within the family. |
| LM-Critic | The knowledge of the genetic risk was to be shared within the ~~family~~ families. |
| Our method | The knowledge of the genetic risk was to be shared within the family. |

Table 5: Examples comparing our system to GPT4 and LM-Critic. Both GPT4 and LM-Critic tend to make unnecessary edits by adding articles or changing pronouns or noun number.

not. The advantage of our system over LM-Critic could be attributed to two components: a better initial fixer which corrects more errors, and a better critic which assesses sentence grammaticality more precisely, as illustrated in Table 2 and Table 4.

## 5 Experiments on Chinese GEC

### 5.1 Data and Model Configuration

We generate 10 million synthetic sentence pairs using 10 million monolingual sentences crawled from the Toutiao website[6]. We train the Chinese BART-large model (Shao et al., 2021) on this data to create the initial fixer $f_0$. To build the monolingual dataset $D_m$, we randomly select 4 million sentences from the CCMatrix corpus (Schwenk et al., 2021), Chinese Lang8 (Zhao et al., 2018), and HSK (Zhang, 2009). For both Lang8 and HSK datasets, we only take the sentences from the source side. When creating the critic, we use the HSK dataset as $D'_m$ and use RoBERTa-wwm-ext (Cui et al., 2020) as our classifier model.

We evaluate the performance of our Chinese GEC system on the NLPCC-2018 test set with the MaxMatch scorer. Following Cao et al. (2021), we use the one-tailed sign test with bootstrap resampling to carry out statistical significance tests.

### 5.2 Results

Since no unsupervised results are available for Chinese GEC, we compare our model with existing supervised models on the NLPCC-2018 test set. Ta-

---
[6]https://www.toutiao.com/

| System | Ens | NLPCC-2018 | | |
|---|---|---|---|---|
| | | P | R | $F_{0.5}$ |
| Zhao and Wang (2020) | ✗ | 44.4 | 22.2 | 37.0 |
| Sun et al. (2022) | ✗ | 46.0 | 27.8 | 40.7 |
| Wu and Wu (2022) | ✗ | 50.6 | 25.2 | 42.1 |
| Zhang et al. (2022b) | ✗ | 45.0 | 33.0 | 45.3 |
| Zhang et al. (2022a) | ✓ | **59.4** | 24.2 | 46.0 |
| Our Unsupervised System | | | | |
| Initial fixer | ✗ | 46.5 | 25.6 | 39.9 |
| + supervised data | ✗ | 55.5 | 29.7 | 47.3 |
| 1st iteration | ✗ | 51.2 | 25.3 | 42.5 |
| 2nd iteration | ✗ | *52.1* | *28.2* | *44.7* |
| + supervised data | ✗ | 57.1 | **28.9** | **47.8**# |

Table 6: Performance (in %) of GEC systems on the NLPCC-2018 test set. Ens: represents if the system uses the ensemble method. The best unsupervised result and the supervised result are shown in ***bold*** and **bold**, respectively. Statistically significant improvements ($p < 0.01$) over initial fixer + supervised data is marked #.

ble 6 shows that our model achieves 44.7 $F_{0.5}$ score, surpassing Wu and Wu (2022) and Sun et al. (2022). It is only 0.6 points below the best-performing supervised single system. When we further fine-tune our unsupervised GEC system with labeled data, the combination of the Chinese Lang8 dataset, and the HSK dataset, our system achieves 47.8 $F_{0.5}$ score, setting a new SOTA on NLPCC-2018. It demonstrates that our unsupervised model can serve as a strong initial checkpoint for supervised training.

# 6 Conclusion

In this paper, we present innovative unsupervised techniques to produce synthetic parallel data and train a critic to evaluate the grammaticality of sentences. By combining our methods with BIFI, we develop an unsupervised GEC system that achieves results comparable to models utilizing substantial labeled data. The core idea is to employ language-independent erroneous models to construct realistic synthetic data, and then create an unsupervised critic utilizing high-confidence predictions from the fixer model. Our system does not require any manually defined or extracted confusion sets, making it an ideal solution for developing GEC models for low-resource languages.

# 7 Limitations

We identified and utilized error patterns in both English and Chinese labeled corpora. While we believe such patterns are language-agnostic, we have not explored their application to other low-resource languages. Future research may delve

further into this area. Additionally, we trained our models using extensive GPU resources, up to 32 A100 GPUs, though similar results can be achieved with just 8 V100 GPUs.

# Acknowledgements

We thank the anonymous reviewers for their helpful comments. This research is supported by a research grant from TikTok (WBS No. A-8000972-00-00). The computational work for this article was partially performed on resources of the National Supercomputing Centre, Singapore (https://www.nscc.sg).

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

# A  Appendix

## A.1  Details on Exploiting Error Patterns

**Validation set selection.** We carry out error pattern analysis on the validation set. Specifically, we follow previous work (Cao et al., 2023; Wu and Wu, 2022) to use BEA-2019 dev set (Bryant et al., 2019) and randomly sample 5,000 sentences from the NLPCC-2018 training set (Zhao et al., 2018) as the validation set for English and Chinese, respectively.

**Vocabulary creation.** We derive the vocabulary from the C4 corpus (Raffel et al., 2020) and the UN Parallel Corpus v1.0 (Ziemski et al., 2016) for English and Chinese, respectively.

**Error type creation.** We use the ERRANT toolkit[7] to extract edits. Specifically, we use the 'all-split' configuration, which merges nothing, when extracting edit pairs from the labeled data. In this way, both the target side and the source side of an edit pair contain at most one token. If the source side of an edit pair is empty, the edit is categorized as an insertion error. If the target side of an edit pair is empty, the edit is categorized as a deletion error. For the rest of the cases, the edit is categorized as a replacement error.

**Complete figures.** We show the insertion and deletion error pattern for English in Figure 7. The insertion and deletion error pattern for Chinese is shown in Figure 8. The replacement error pattern for English is shown in Figure 9. The replacement error pattern for Chinese is shown in Figure 10.

## A.2  Extracting GED Pseudo-Labels from the Fixer

The complete correlation between the probability of producing $\hat{y}^{(i)}$ and precision of $z^{(i)}$ is shown in Figure 11.

## A.3  Detailed Experimental Settings

**Implementation details and training configuration.**

We build our fixer using both the fairseq[8] and transformers[9] toolkit. Specifically, since the Flan-T5-xxl model has around 11B parameters, we use the transformers toolkit with DeepSpeed[10] ZeRO-Offload to build the fixer for English and use the fairseq toolkit to build the rest of the components. For English GEC, we use 32 NVIDIA A100 GPUs. For Chinese GEC, we use 8 NVIDIA A100 GPUs. The experiments took 14 days for English and 2 days in total for Chinese. We use the default training configuration under different toolkits unless otherwise stated. The detailed training configurations for English and Chinese are shown in Table 8 and Table 9, respectively. The best checkpoint is selected based on the performance on the validation set. Specifically, when building the fixer, we follow Yasunaga and Liang (2021) to randomly sample 5,000 sentences from the obtained training

---

[7]https://github.com/chrisjbryant/errant
[8]https://github.com/facebookresearch/fairseq
[9]https://github.com/huggingface/transformers
[10]https://github.com/microsoft/DeepSpeed

sentence pairs as the validation data for both English and Chinese. When building the critic, we follow the approach used by Yasunaga et al. (2021) to randomly select 600 grammatical sentences and 600 ungrammatical sentences from the BEA-2019 dev set and Chinese Lang8 dataset as the validation set for English and Chinese, respectively.

**Hyper-parameter settings.** We tune two hyper-parameters in our system, the edit distance threshold, as mentioned in §3.2.2, and the masking percentage, denoted as $p\%$, which is outlined in §3.3. We select the edit distance threshold from {1, 2, 3, 4, 5} for English GEC and select the the edit distance threshold from {0, 1, 2} for Chinese. For both English and Chinese $p$ is selected from {5, 10, 15}. For English, the edit distance threshold 2 and $p$ equals 5% give the best performance on the validation set. For Chinese, the edit distance threshold 1 and $p\%$ equals 10% give the best performance on the validation set.

**Parameters for synthetic data generation.** Table 10 shows the parameter values used when generating the synthetic data. Note that these values are set to mimic the error distribution in real erroneous corpora.

## A.4 Experiments on German and Russian

We use German (Falko-MERLIN dataset) and Russian (RULEC-GEC dataset) to demonstrate our method's performance in additional languages.

For both languages, we use mT5-xxl instead of Flan-T5-xxl as the base model and generate 10 million synthetic sentence pairs by corrupting the sentences from UN-Corpus v1.0. Following the setup in Section 4.1 and Section 5.1, we randomly collect 10 million sentences from the CCMatrix (Schwenk et al., 2021) corpus, Falko-MERLIN (Boyd et al., 2014) dataset, and cLang8(Rothe et al., 2021) dataset for German. For both Falko-MERLIN dataset and cLang8 dataset, we take the sentences from the source side (not annotated sentences), which could be grammatical or ungrammatical. We randomly collect 10 million sentences from the CCMatrix (Schwenk et al., 2021) corpus, RULEC-GEC (Rozovskaya and Roth, 2019) dataset, and cLang8 (Rothe et al., 2021) dataset for Russian. For both RULEC-GEC dataset and cLang8 dataset, we also take the sentences from the source side. The results are shown in the Table 7. Note that no unsupervised baselines exist in German and Russian GEC.

| System | Falko-MERLIN (P/R/$F_{0.5}$) | RULEC-GEC (P/R/$F_{0.5}$) |
|---|---|---|
| Our Unsupervised System | | |
| Initial fixer | 74.3/50.1/67.8 | 55.8/22.0/42.6 |
| 1st iteration | 76.2/64.2/73.4 | 60.1/27.7/48.7 |
| 2nd iteration | 76.5/67.8/74.5 | 60.4/30.1/50.3 |
| Supervised SOTA System | | |
| Rothe et al. (2021) | -/-/76.0 | -/-/51.6 |
| Sorokin (2022) | -/-/- | 73.7/27.3/55.0 |

Table 7: Performance (in %) of GEC systems on the Falko-MERLIN and RULEC-GEC test sets.

| Configuration | Value |
|---|---|
| | Fixer |
| Devices | 32 NVIDIA A100 GPU |
| Batch Size per GPU | 256 |
| Update Frequency | 1 |
| Loss function | label smoothed cross entropy (label-smoothing=0.1) |
| Model architecture | Flan-T5-xxl |
| Optimizer | Adamw ($\beta_1 = 0.9$, $\beta_2 = 0.999$, $\epsilon = 1 \times 10^{-8}$) |
| Learning rate | $2.00 \times 10^{-5}$ |
| Learning rate scheduler | Linear |
| Warmup | 0 |
| Number of epochs | 10 |
| | Critic |
| Devices | 1 NVIDIA A100 GPU |
| Batch Size per GPU | 10000 |
| Update Frequency | 1 |
| Loss function | cross entropy |
| Model architecture | RoBERTa-base |
| Optimizer | Adam ($\beta_1 = 0.9$, $\beta_2 = 0.98$, $\epsilon = 1 \times 10^{-6}$) |
| Learning rate | $1.00 \times 10^{-5}$ |
| Learning rate scheduler | Polynomial decay |
| Warmup | 400 |
| Number of epochs | 40 |

Table 8: Experimental configuration on English.

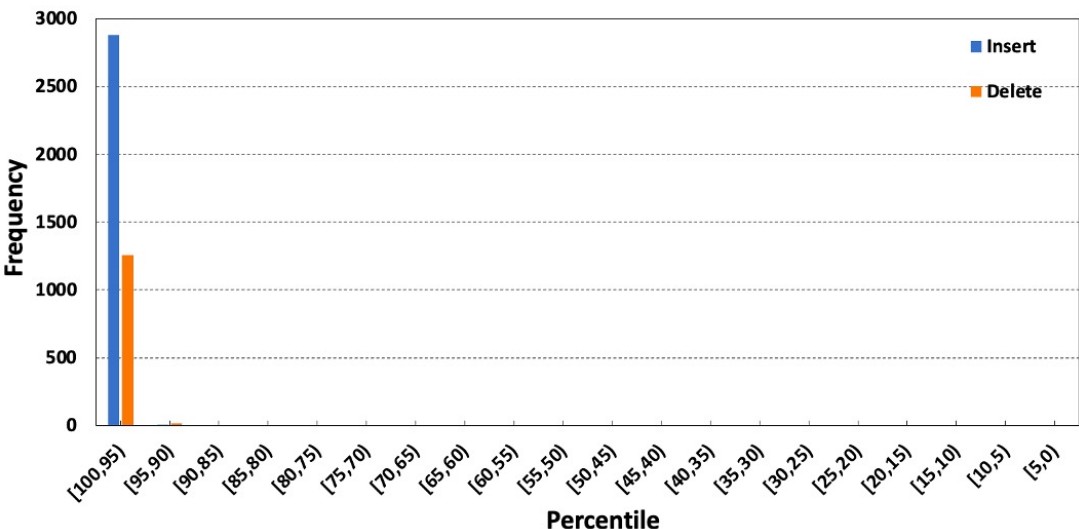

Figure 7: The erroneous token distribution for insertion and deletion errors for English. The tokens in the vocabulary are ordered by decreasing frequency.

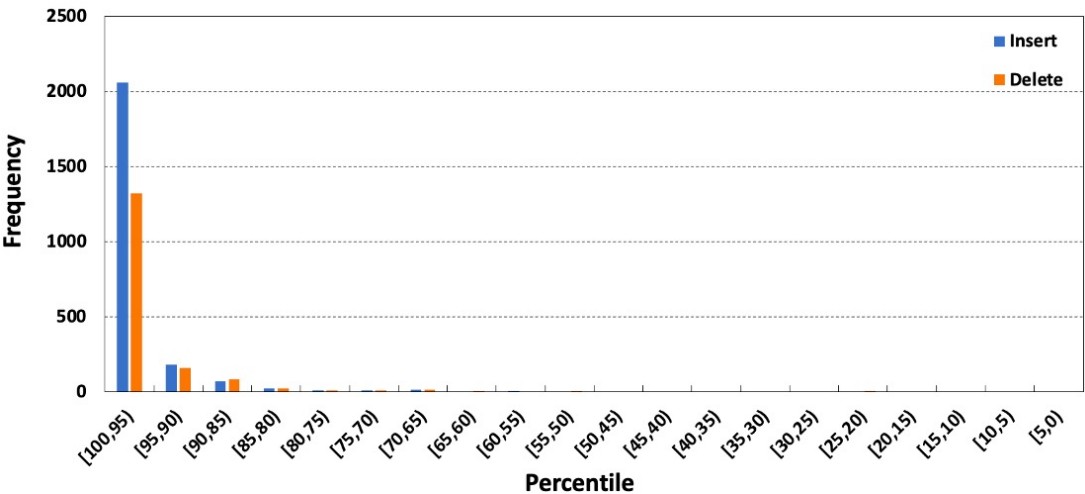

Figure 8: The erroneous token distribution for insertion and deletion errors for Chinese. The tokens in the vocabulary are ordered by decreasing frequency.

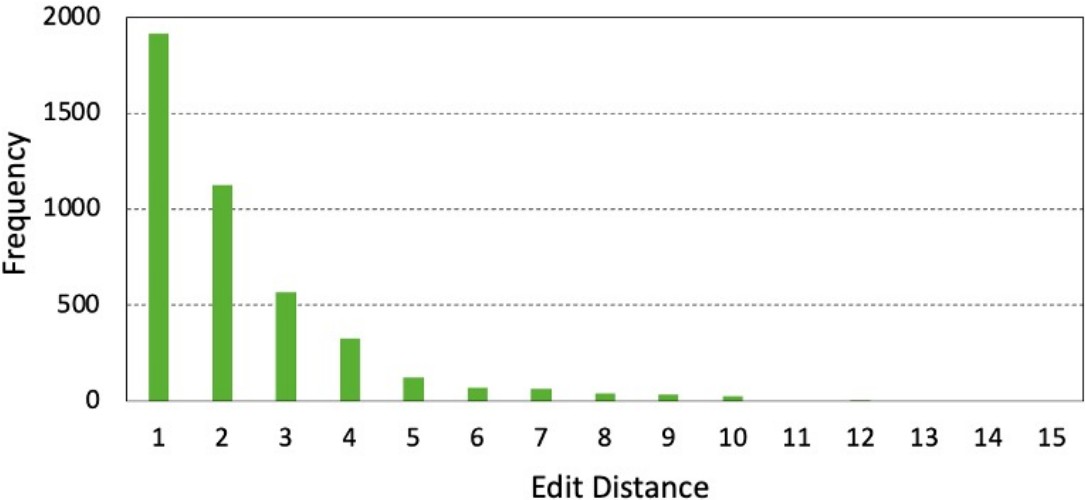

Figure 9: The character-level edit distance between an erroneous token and its corresponding target token for replacement error for English.

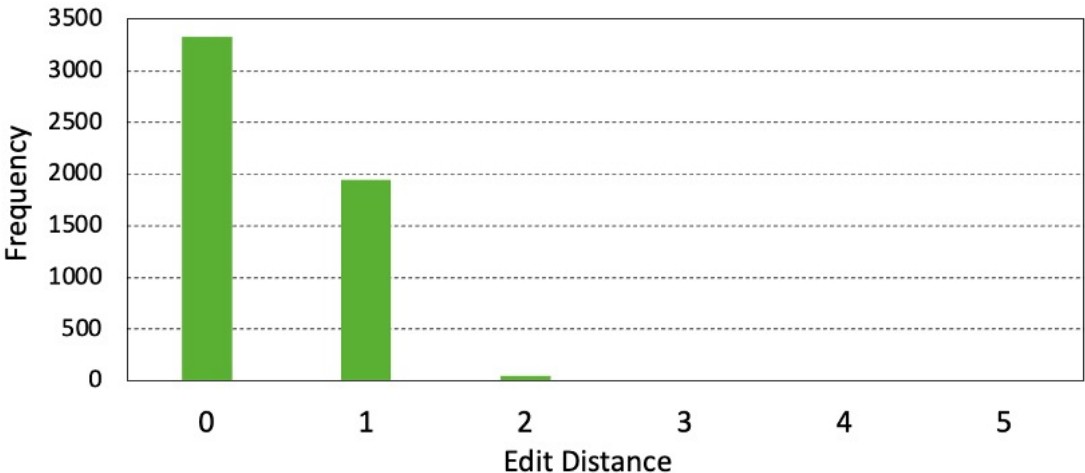

Figure 10: The character-level edit distance between an erroneous token and its corresponding target token for replacement error for Chinese.

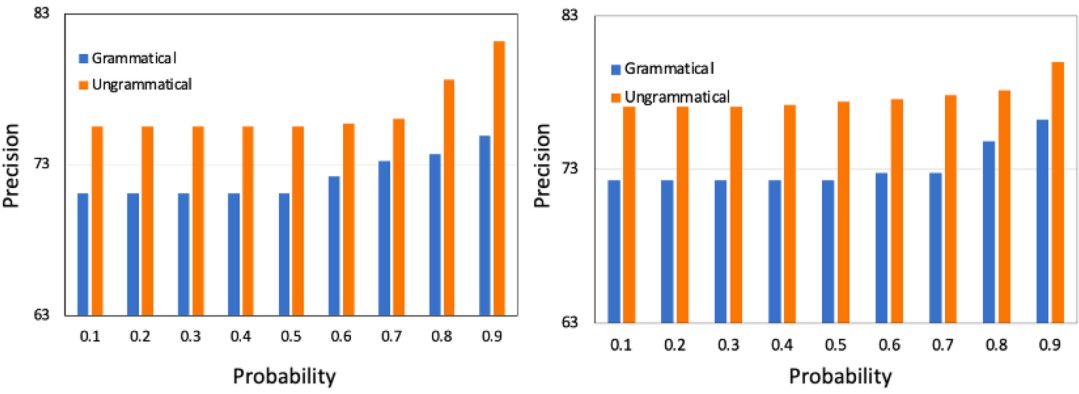

Figure 11: Correlation between the probability of producing $\hat{y}^{(i)}$ and precision of $z^{(i)}$. *Left:* English; *Right:* Chinese.

| Configuration | Value |
|---|---|
| | Fixer |
| Devices | 8 NVIDIA A100 GPU |
| Max tokens per GPU | 7000 |
| Update Frequency | 1 |
| Loss function | label smoothed cross entropy (label-smoothing=0.1) |
| Model architecture | Chinese BART-large |
| Optimizer | Adam ($\beta_1 = 0.9$, $\beta_2 = 0.98$, $\epsilon = 1 \times 10^{-6}$) |
| Learning rate | $1.00 \times 10^{-5}$ |
| Learning rate scheduler | Polynomial decay |
| Warmup | 0 |
| Number of epochs | 15 |
| | Critic |
| Devices | 1 NVIDIA A100 GPU |
| Max tokens per GPU | 10000 |
| Update Frequency | 1 |
| Loss function | cross entropy |
| Model architecture | RoBERTa-wwm-ext |
| Optimizer | Adam ($\beta_1 = 0.9$, $\beta_2 = 0.98$, $\epsilon = 1 \times 10^{-8}$) |
| Learning rate | $1.00 \times 10^{-5}$ |
| Learning rate scheduler | Polynomial decay |
| Warmup | 400 |
| Number of epochs | 40 |

Table 9: Experimental configuration on Chinese.

| | English | Chinese |
|---|---|---|
| $p_{del}$ | 0.15 | 0.15 |
| $p_{ins}$ | 0.35 | 0.35 |
| $p_{rep}$ | 0.50 | 0.50 |
| error count distribution | multinoulli (0.05, 0.07, 0.25, 0.35, 0.28) | multinoulli (0.01, 0.32, 0.29, 0.20, 0.18) |
| $LIST_{ID}$ | [5, 10, 40, 80, 200, 500, 1000, 2800] | [35, 95, 187, 274, 372, 561, 787, 1176, 1995] |

Table 10: Parameters for synthetic data generation.