# OpenReview forum: "Unsupervised Grammatical Error Correction Rivaling Supervised Methods"
_EMNLP/2023/Conference — EMNLP 2023 Main_

### Official Review · Reviewer_Y4qz · 2023-07-20

**Typos Grammar Style And Presentation Improvements:** 1. line 126
**Soundness:** 4

**Excitement:**

3: Ambivalent: It has merits (e.g., it reports state-of-the-art results, the idea is nice), but there are key weaknesses (e.g., it describes incremental work), and it can significantly benefit from another round of revision. However, I won't object to accepting it if my co-reviewers champion it.

**Paper Topic And Main Contributions:**

This paper proposes an unsupervised method for grammatical error correction based on the Break-It-Fix-It method.

The primary contributions of this study lie in the enhancements made to both the 'fixer' and the 'critic' components within the BIFI framework.

For the fixer, this paper puts forward a simple data augmentation method based on MLM and heuristic rules. For the critic, this paper proposes to select high-confidence samples from the fixer's predictions and use them for training. During training, they also use two simple tricks to further improve performance, i.e., masked data augmentation and self-knowledge distillation.

The results show their unsupervised method significantly surpasses previous methods, and can further enhance model performance under the supervised setting.

**Questions For The Authors:**

1. In Table 1, can you provide the performance of using T5 with different model sizes (e.g., T5-base or T5-large)?
2. In Figure 5, can you offer any insightful explanations about why the performance of your fixer continuously increases while other counterparts don't?

**Reasons To Accept:**

1. SoTA performance on both English and Chinese benchmarks.
2. Clear paper writing.
3. Reasonable expanded analyses to support the motivation of each component.

**Reasons To Reject:**

1. The method the authors proposed seems to be a straightforward extension of the LM-Critic work (Yasunaga et al., 2021). The heuristic rules for generating data for training the initial fixer, the self-knowledge distillation when training the critic, the masked data augmentation when training the critic, and selecting high-confidence pseudo labels when training the critic, collectively form a set of useful techniques. However, in my view, these strategies do not represent a significant novelty.
2. Despite achieving impressive performance, this work relies heavily on a large backbone model and extensive multi-task pretraining, i.e., Flan-T5-11B. Moreover, the method only yields a modest improvement of approximately 0.5 F score under the supervised setting on both English and Chinese benchmarks. The authors should conduct additional experiments under low-resource settings to further validate the effectiveness of their methods. This could include testing on low-resource languages and smaller backbone models.

**Reproducibility:**

4: Could mostly reproduce the results, but there may be some variation because of sample variance or minor variations in their interpretation of the protocol or method.

**Reviewer Confidence:**

4: Quite sure. I tried to check the important points carefully. It's unlikely, though conceivable, that I missed something that should affect my ratings.

---

> ### Author Rebuttal · Authors · 2023-08-29
>
> Thank you very much for the review suggestions. Here is our response to the “reasons to reject” and to the questions.
>
> For the novelty of the paper, we want to clarify that we have made 4 substantial innovative contributions beyond a “straightforward extension of LM-Critic”:
> 1. Although both LM-Critic and our method require a fixer and a critic, we define the critic in a completely different way. Different from LM-Critic which defines the critic from GPT2 and a confusion set, we train and refine the critic from the fixer. This design has two advantages. First, our method doesn’t rely on the confusion set, making it fully unsupervised. Second, the fixer and the critic can iteratively improve each other. As illustrated in Figure 5, this further improves the fixer’s eventual performance.
> 2. Our synthetic data generation method includes some important new techniques. For example, in using masked language models (MLM) to generate synthetic data, Sun et al. (2022) present a similar idea, but we are the first to propose to use edit-distance information and high-frequency tokens to filter the generated errors. As shown in Table 2, this simple technique has been proven to be extremely important to the fixer’s performance.
> 3. Our paper is the first to demonstrate that scaling up the backbone model size (BART-base -> T5-large (we provide this model’s performance below) -> Flan-T5-xxl) can greatly improve unsupervised performance. This is non-trivial, because in the supervised GEC setting, people have found that the model size is not that important. For example, comparing Zhang et al. (2022) (BART-large) and Rother et al. (2021) (T5-xxl), the performance only differs by 1.9 points on the CoNLL-2014 test set. On the other hand, we have shown in Table 1 that our method significantly outperforms ChatGPT and GPT4, showing that good results not only require a big backbone model, but also a carefully designed training method proposed by us.
> 4. Our method is the first unsupervised method that rivals the performance of supervised SOTA systems, not only in high-resource languages (English, Chinese), but also in low-resource languages (German and Russian). German and Russian have been widely used to evaluate low-resource performance in GEC. Their performance is shown below.
>
>
> The performance of T5-large using our unsupervised method is shown here:
> | | CoNLL-2014 (P/R/F0.5) | BEA-2019 (P/R/F0.5) |
> |-----------|-----------|-----------|
> | Pretrain | 67.3/38.5/58.5 | 71.3/45.2/63.9 |
> | Stage1 | 70.3/44.3/62.9 | 74.4/55.3/69.6 |
> | Stage2 | 72.3/46.0/64.8 | 76.6/56.3/71.4 |
> | Stage3 | 72.0/50.4/66.3 | 75.5/61.6/72.3 |
>
>
> The unsupervised performance on German and Russian is shown here:
> For both German (Falko-MERLIN dataset) and Russian (RULEC-GEC dataset) we use mT5-xxl instead of Flan-T5-xxl as the base model and generated 10 million synthetic sentence pairs by corrupting the sentences from UN-Corpus v1.0. Following the setup in Section 4.1 and Section 5.1, we randomly collect 10 million sentences from the CCMatrix corpus, Falko-MERLIN dataset, and clang8 dataset for German. For both Falko-MERLIN dataset and clang8 dataset ,we take the sentence from the source side (not annotated sentences), which could be grammatical or ungrammatical. We randomly collect 10 million sentences from the CCMatrix corpus, RULEC-GEC dataset, and clang8 dataset for Russian. For both RULEC-GEC dataset, and clang8 dataset we also take the sentence from the source side. The results are shown in the table below (No unsupervised baselines exist in the German and Russian GEC).
> | | German (P/R/F0.5) | Russian (P/R/F0.5) |
> |-----------|-----------|-----------|
> |Our Unsupervised System|
> | Pretrain | 74.3/50.1/67.8 | 55.8/22.0/42.6 |
> | Stage1 | 76.2/64.2/73.4 | 60.1/27.7/48.7 |
> | Stage2 | 76.5/67.8/74.5 | 60.4/30.1/50.3 |
> |Supervised SOTA Systems|
> | gT5-xxl (Rothe et al., 2021) | -/-/76.0 | -/-/51.6 |
> | EditScorer (Sorokin, 2022) without editscorer | -/-/- | 65.7/27.4/51.3 |
>
> This result demonstrates that our method works well on low-resource languages. In particular, our unsupervised F0.5 scores are in close proximity to strong supervised results (Rothe et al., 2021, Sorokin, 2022). The highest model for Russian GEC is (Sorokin, 2022), which proposed an editscorer on top of the GEC model. In the table below, we demonstrate that when we combine our unsupervised model with the editscorer, we can get a comparable performance.
> | | Russian (P/R/F0.5) |
> |-----------|-----------|
> |Our Unsupervised System|
> | Stage2 | 60.4/30.1/50.3 |
> |   +editscorer | 70.3/28.9/54.6 |
> |Supervised SOTA Systems|
> | EditScorer (Sorokin, 2022) with editscorer | 73.7/27.3/55.0 |
>
> These results demonstrate that our unsupervised system compares favorably with strong supervised SOTA systems on low-resource languages.
>
>
> The reviewer questions whether our method works well for low-resource languages. Our results in the above tables demonstrate that our unsupervised system is competitive with strong supervised SOTA GEC systems in German and Russian (two representative low-resource languages for GEC).
>
>
> The reviewer also questions whether our method works well for small backbone models. As shown in Table 1 (also line 385), we have used BART-base (220 million parameters) as one of the smaller backbone models to compare our method against LM-Critic. The result shows that our method is 5.3 points and 2.1 points higher than the LM-Critic on the CoNLL-2014 and BEA-2019 test set, respectively. When comparing against humans using the method proposed by Bryant and Ng, (2015), our BART-base model (73.0 F0.5 score) also surpassed human performance (72.5 F0.5 score). This makes ours the first unsupervised system that surpasses human performance. We will add this comparison to the revision.
>
> In the following table, we show the unsupervised system’s performance based on T5-large (a smaller backbone).
> | | CoNLL-2014 (P/R/F0.5) | BEA-2019 (P/R/F0.5) |
> |-----------|-----------|-----------|
> | Pretrain | 67.3/38.5/58.5 | 71.3/45.2/63.9 |
> | Stage1 | 70.3/44.3/62.9 | 74.4/55.3/69.6 |
> | Stage2 | 72.3/46.0/64.8 | 76.6/56.3/71.4 |
> | Stage3 | 72.0/50.4/66.3 | 75.5/61.6/72.3 |
>
> The competitive results from T5-large also shows that our method works well under the small backbone model setting.
>
> Questions:
> 1.   Yes. Our response to the second reason to reject has provided the performance of the T5-large model.
> 2.   Yes. Figure 5 shows that the fixer’s performance only marginally improves using either LM-Critic or our critic w/o critic update. This is because the critic is fixed throughout the iteration. Once the critic is not updated, the paired data extracted by the critic in the subsequent iterations may not further benefit the fixer’s training since no extra knowledge has been injected into the system. As a result, the fixer is learning from itself, unintentionally causing the over-fitting problem. This phenomenon can be broken by adding masking-based data augmentation methods and updating the critic throughout the iteration, as shown in Figure 5 and Figure 6.
>
> Typos:
>
> Thanks for the suggestion regarding these typos, we will update accordingly. For the third typo, it should be our critic, as labeled in Figure 5. This is because Figure 5 shows the performance of the fixer under different types of critic.
>
> Reference:
>
> Christopher Bryant, and Hwee Tou Ng. How Far are We from Fully Automatic High Quality Grammatical Error Correction? In ACL 2015.
>
> Alexey Sorokin. Improved grammatical error correction by ranking elementary edits. In EMNLP 2022.
>
> Sascha Rothe, Jonathan Mallinson, Eric Malmi, Sebastian Krause, and Aliaksei Severyn. 2021. A simple recipe for multilingual grammatical error correction. In ACL 2021.

---

### Official Review · Reviewer_fd3a · 2023-08-05

**Soundness:** 4

**Excitement:**

3: Ambivalent: It has merits (e.g., it reports state-of-the-art results, the idea is nice), but there are key weaknesses (e.g., it describes incremental work), and it can significantly benefit from another round of revision. However, I won't object to accepting it if my co-reviewers champion it.

**Paper Topic And Main Contributions:**

This paper presents an unsupervised pipeline for training a grammatical error correction (GEC) system using unlabeled data. In comparison to the LM-Critic framework, the primary contribution of this work lies in the integration of several techniques for synthetic data generation into the existing framework, thereby taking a step towards eliminating the reliance on external sources of labeled data. Specifically, the proposed method leverages the empirical distribution of error patterns extracted from the developmental data to corrupt the data through three token-level edit operations, including replacement, deletion, and insertion. A masked language model (MLM) is employed in this process to handle token replacement. Additionally, the paper introduces self-knowledge distillation in the LM-Critic training loop to enhance the model's generalization ability. Based on the experiments conducted, the new training pipeline outperforms the unsupervised baselines, as well as the vanilla LM-Critic pipeline.

**Questions For The Authors:**

- A. Table 3 and 5 present some intuitive cases produced by different methods. How were these examples selected? Are they representative enough of the behavior of each method?
- B. Line 298: The [MASK] token is used to introduce noise in the self-training process. Is it different from other specific tokens regarding injecting noise?
- C. Line 312: $f_c$ represents the sum over soft frequencies for class $c$, but in the previous section, $f$ is used to indicate the fixer. In my understanding, $f_c$ should be unrelated to the fixer since it is conditional on the critic model parameter. Did you misuse the $f$ symbol?

**Reasons To Accept:**

- **Performance**: The proposed method demonstrates superior performance compared to the fundamental LM-Critic approach. The experimental results are promising, showing comparable performance to supervised methods, which highlights the effectiveness of the new synthetic data generation and training techniques.
- **Motivation**: The motivation behind adopting most components of the fixer-critic training loop from the mature LM-Critic framework is sound. Additionally, the new perturbation method based on empirical observations of error patterns adds credibility to the proposed approach.

**Reasons To Reject:**

- **Novelty**: While the experimental results are encouraging, the high-level design of the LM-Critic framework remains largely unchanged. The primary contribution lies in providing improved synthetic data generation methods for training the fixer and critic, limiting the novelty of the paper.
- **Scalability**: The empirical observation of error patterns may pose challenges when dealing with new languages and benchmarks. Analyzing error patterns could result in additional development costs and may not be easily scalable to other contexts.
- **Robustness**: There is a concern about the vulnerability of the proposed generation method if the validation data does not highly represent the error patterns of other data splits. This could impact the robustness of the system in different scenarios.

**Update:** I increased my soundness score after reading the reply. Please see my follow-up comment below.

**Reproducibility:**

4: Could mostly reproduce the results, but there may be some variation because of sample variance or minor variations in their interpretation of the protocol or method.

**Reviewer Confidence:**

4: Quite sure. I tried to check the important points carefully. It's unlikely, though conceivable, that I missed something that should affect my ratings.

**Typos Grammar Style And Presentation Improvements:**

- Line 280: In Figure $4^4$ -> In Figure 4
- Line 394: clang8 -> cLang8
- Line 433: Table 4.6. -> Table 4.

---

> ### Author Rebuttal · Authors · 2023-08-29
>
> Thank you very much for the review suggestions. Here is our response to the “reasons to reject” and to the questions.
>
> Novelty: we want to clarify that our method has four innovative contributions:
> 1. Although both LM-Critic and our method require a fixer and a critic, we define the critic in a completely different way. Different from LM-Critic which defines the critic from GPT2 and a confusion set, we train and refine the critic from the fixer. This design has two advantages. First, our method doesn’t rely on the confusion set, making it fully unsupervised. Second, the fixer and the critic can iteratively improve each other. As illustrated in Figure 5, this further improves the fixer’s eventual performance.
> 2. Our synthetic data generation method includes some important new techniques. For example, in using masked language models (MLM) to generate synthetic data, Sun et al. (2022) present a similar idea, but we are the first to propose to use edit-distance information and high-frequency tokens to filter the generated errors. As shown in Table 2, this simple technique has been proven to be extremely important to the fixer’s performance.
> 3. Our paper is the first to demonstrate that scaling up the backbone model size (BART-base -> T5-large (we provide this model’s performance below) -> Flan-T5-xxl) can greatly improve unsupervised performance. This is non-trivial, because in the supervised GEC setting, people have found that the model size is not that important. For example, comparing Zhang et al. (2022) (BART-large) and Rother et al. (2021) (T5-xxl), the performance only differs by 1.9 points on the CoNLL-2014 test set. On the other hand, we have shown in Table 1 that our method significantly outperforms ChatGPT and GPT4, showing that good results not only require a big backbone model, but also a carefully designed training method proposed by us.
> 4. Our method is the first unsupervised method that rivals the performance of supervised SOTA systems, not only in high-resource languages (English, Chinese), but also in low-resource languages (German and Russian). German and Russian have been widely used to evaluate low-resource languages’ performance in GEC. Their performance is shown below.
>
> The performance of T5-large using our unsupervised method is shown here:
>
> | | CoNLL-2014 (P/R/F0.5) | BEA-2019 (P/R/F0.5) |
> |-----------|-----------|-----------|
> | Pretrain | 67.3/38.5/58.5 | 71.3/45.2/63.9 |
> | Stage1 | 70.3/44.3/62.9 | 74.4/55.3/69.6 |
> | Stage2 | 72.3/46.0/64.8 | 76.6/56.3/71.4 |
> | Stage3 | 72.0/50.4/66.3 | 75.5/61.6/72.3 |
>
> The unsupervised performance on German and Russian is shown here:
>
> We use German (Falko-MERLIN dataset) and Russian (RULEC-GEC dataset) to demonstrate our model's performance in low-resource languages. For both languages, we use mT5-xxl instead of Flan-T5-xxl as the base model and generated 10 million synthetic sentence pairs by corrupting the sentences from UN-Corpus v1.0. Following the setup in Section 4.1 and Section 5.1, we randomly collect 10 million sentences from the CCMatrix corpus, Falko-MERLIN dataset, and clang8 dataset for German. For both Falko-MERLIN dataset and clang8 dataset, we take the sentence from the source side (not annotated sentences), which could be grammatical or ungrammatical. We randomly collect 10 million sentences from the CCMatrix corpus, RULEC-GEC dataset, and clang8 dataset for Russian. For both RULEC-GEC dataset, and clang8 dataset we also take the sentence from the source side. The results are shown in the table below (No unsupervised baselines exist in the German and Russian GEC).
> | | German (P/R/F0.5) | Russian (P/R/F0.5) |
> |-----------|-----------|-----------|
> |Our Unsupervised System|
> | Pretrain | 74.3/50.1/67.8 | 55.8/22.0/42.6 |
> | Stage1 | 76.2/64.2/73.4 | 60.1/27.7/48.7 |
> | Stage2 | 76.5/67.8/74.5 | 60.4/30.1/50.3 |
> |Supervised SOTA Systems|
> | gT5-xxl (Rothe et al., 2021) | -/-/76.0 | -/-/51.6 |
> | EditScorer (Sorokin, 2022) without editscorer | -/-/- | 65.7/27.4/51.3 |
>
> This result demonstrates that our method works well on low-resource languages. In particular, our unsupervised F0.5 scores are in close proximity to strong supervised results (Rothe et al., 2021, Sorokin, 2022). The highest model for Russian GEC is (Sorokin, 2022), which proposed an editscorer on top of the GEC model. In the table below, we demonstrate that when we combine our unsupervised model with the editscorer, we can get a comparable performance.
> | | Russian (P/R/F0.5) |
> |-----------|-----------|
> |Our Unsupervised System|
> | Stage2 | 60.4/30.1/50.3 |
> |   +editscorer | 70.3/28.9/54.6 |
> |Supervised SOTA Systems|
> | EditScorer (Sorokin, 2022) with editscorer | 73.7/27.3/55.0 |
>
> These results demonstrate that our unsupervised system compares favorably with strong supervised SOTA systems on low-resource languages.
>
>
> Scalability & Robustness: the reviewer questions whether the methods and the error patterns proposed in the paper work across different domains and languages. Here we present some additional analysis to prove that it is true.
>
> First, we test the two error patterns proposed in the paper:
> 1. Most deletion and insertion errors occur within 95th percentile of the vocabulary.
> 2. Most replacement errors have small (<5) edit distance.
>
> We analyze three additional commonly used English GEC evaluation datasets across different domains: JFLEG, CoNLL-2013, and the CWEB dataset. In the table below, we show how much of the erroneous token lies within the first 95 percentile of the vocabulary for deletion and insertion errors. We also show how much percentage of the edit pairs has an edit distance less than 5 for the replacement error. The result matches our proposed patterns.
> | | Deletion & Insertion | Replacement |
> |-----------|-----------|-----------|
> | JFLEG | 93% | 90% |
> | CoNLL-2013 | 90% | 88% |
> | CWEB | 96% | 94% |
>
> We now analyze two representative low-resource languages in GEC: German (Falko-MERLIN dataset) and Russian (RULEC-GEC dataset). The table below shows how much of an erroneous token lies within the top 95 percentile of the vocabulary for deletion and insertion errors, as well as the percentage of edit pairs with an edit distance of less than 5 for replacement errors. The results reveal patterns similar to those observed in English and Chinese.
> | | Deletion & Insertion | Replacement |
> |-----------|-----------|-----------|
> | German | 89% | 88% |
> | Russian| 90% | 89% |
>
> Our synthetic data generation method also shows competitive performance against related works in German and Russian. Following the setup in Section 4.3, we compare our synthetic data generation method with Sun et al. (2022) and Grundkiewicz and Junczys-Dowmunt. (2019). The results are shown in the following table. It shows that our synthetic data generation method works well for low-resource languages and outperforms existing methods.
> | | Deletion & Insertion | Replacement |
> |-----------|-----------|-----------|
> | Spellchecker (Grundkiewicz and Junczys-Dowmunt. (2019)) | 24.8/13.7/21.3 | 15.4/11.1/14.2 |
> | Translation pair (Sun et al. (2022)) | 28.8/12.4/22.8 | 16.7/11.4/15.2 |
> | Ours| 32.0/14.0/25.5 | 19.8/13.8/18.2 |
>
> Finally, we test the F0.5 scores of the unsupervised GEC system on German (Falko-MERLIN dataset) and Russian (RULEC-GEC dataset) end-to-end. The results shown in the response for “Novelty” demonstrate that our unsupervised system compares favorably with strong supervised SOTA systems on low-resource languages.
>
> All these analyses show that our method is scalable, robust, and works effectively across different languages.
>
>
> Questions:
> 1. The cases presented in Table 3 and Table 5 are randomly selected. Due to space limitation, we only show a few representative examples produced by different systems.
> 2. In our experiment, we tried using both [MASK] and [UNK] tokens during the self-training process. There is not much difference between the critic’s and fixer’s performance when using either [MASK] or [UNK] token. For consistency consideration, we use the [MASK] token.
> 3. Thanks for pointing out this problem. Yes, we misused the f symbol in line 312, f should be used to indicate the fixer. We will change this symbol in line 312 to avoid the confusion.
>
> Typos:
> Thanks for spotting these problems, we will correct them.
>
> References:
>
> Xin Sun, Tao Ge, Shuming Ma, Jingjing Li, Furu Wei, and Houfeng Wang. A unified strategy for multilingual grammatical error correction with pre-trained cross-lingual language model. In IJCAI 2022.
>
> Roman Grundkiewicz and Marcin Junczys-Dowmunt. Minimally-Augmented Grammatical Error Correction. In W-NUT 2019.
>
> Yue Zhang, Bo Zhang, Zhenghua Li, Zuyi Bao, Chen Li, and Min Zhang. SynGEC: Syntax-enhanced grammatical error correction with a tailored GEC oriented parser. In EMNLP 2022.
>
> Sascha Rothe, Jonathan Mallinson, Eric Malmi, Sebastian Krause, and Aliaksei Severyn. 2021. A simple recipe for multilingual grammatical error correction. In ACL 2021.
>
> Alexey Sorokin. Improved grammatical error correction by ranking elementary edits. In EMNLP 2022.

---

### Official Review · Reviewer_XoJu · 2023-08-05

**Soundness:** 4

**Excitement:**

4: Strong: This paper deepens the understanding of some phenomenon or lowers the barriers to an existing research direction.

**Missing References:**

The best single system on BEA-2019 is (Sorokin, 2022) https://aclanthology.org/2022.emnlp-main.785.pdf, its score is 77.1. Please refer to it in Table 1 and correct the last paragraph of Section 4.2

**Paper Topic And Main Contributions:**

The paper proposes a new method of unsupervised grammatical error correction. In comparison to the previous approach of (Yasunaga et al., 2021) it uses a more natural method of text corruption, thus significantly improving the model quality. When using a Flan-XXL fixer, the proposed model almost reaches the level of supervised models. Using the same fixer that (Yasunaga et al., 2021) it also provides a slight improvement over previous results.

**Questions For The Authors:**

A) I doubt your method will work for low-resource languages since you have no FLAN-XXL for these languages. Even the analogue of BART-base would be of much lower quality or your would have to use mBART.
B) is it true that your method of token replacement works only for the words that consist of single BPE token?
C) I do not understand the intuition behind soft labeling in formula (2), in particular, the squaring of probabilities. Can you briefly explain it inside your work?

**Reasons To Accept:**

* SOTA in unsupervised grammatical error correction for English and Chinese
* a new method that does not utilize any manual effort for creating the unsupervised data is proposed
* good analysis and ablations

**Reasons To Reject:**

No reasons

**Reproducibility:**

3: Could reproduce the results with some difficulty. The settings of parameters are underspecified or subjectively determined; the training/evaluation data are not widely available.

**Reviewer Confidence:**

5: Positive that my evaluation is correct. I read the paper very carefully and I am very familiar with related work.

---

> ### Author Rebuttal · Authors · 2023-08-29
>
> Thank you very much for the positive review. Here is the response to the questions:
>
> Question A:
> We use German (Falko-MERLIN dataset) and Russian (RULEC-GEC dataset) to demonstrate our model's performance in low-resource languages. German and Russian have been widely used to evaluate low-resource languages’ performance in GEC.
>
> For both languages, we use mT5-xxl instead of Flan-T5-xxl as the base model and generated 10 million synthetic sentence pairs by corrupting the sentences from UN-Corpus v1.0. Following the setup in Section 4.1 and Section 5.1, we randomly collect 10 million sentences from the CCMatrix corpus, Falko-MERLIN dataset, and clang8 dataset for German. For both Falko-MERLIN dataset, and clang8 dataset we take the sentence from the source side (not annotated sentences), which could be grammatical or ungrammatical. We randomly collect 10 million sentences from the CCMatrix corpus, RULEC-GEC dataset, and clang8 dataset for Russian. For both RULEC-GEC dataset and clang8 dataset, we also take the sentence from the source side. The results are shown in the table below (No unsupervised baselines exist in the German and Russian GEC).
> | | German (P/R/F0.5) | Russian (P/R/F0.5) |
> |-----------|-----------|-----------|
> |Our Unsupervised System|
> | Pretrain | 74.3/50.1/67.8 | 55.8/22.0/42.6 |
> | Stage1 | 76.2/64.2/73.4 | 60.1/27.7/48.7 |
> | Stage2 | 76.5/67.8/74.5 | 60.4/30.1/50.3 |
> |Supervised SOTA Systems|
> | gT5-xxl (Rothe et al., 2021) | -/-/76.0 | -/-/51.6 |
> | EditScorer (Sorokin, 2022) without editscorer | -/-/- | 65.7/27.4/51.3 |
>
> This result demonstrates that our method works well on low-resource languages. In particular, our unsupervised F0.5 scores are in close proximity to strong supervised results (Rothe et al., 2021, Sorokin, 2022). The best model for Russian GEC is (Sorokin, 2022), which proposed an editscorer on top of the GEC model. In the table below, we demonstrate that when we combine our unsupervised model with the editscorer, we can get a comparable performance.
> | | Russian (P/R/F0.5) |
> |-----------|-----------|
> |Our Unsupervised System|
> | Stage2 | 60.4/30.1/50.3 |
> |   +editscorer | 70.3/28.9/54.6 |
> |Supervised SOTA Systems|
> | EditScorer (Sorokin, 2022) with editscorer | 73.7/27.3/55.0 |
> These results demonstrate that our unsupervised system compares favorably with strong supervised SOTA systems on low-resource languages.
>
> Question B:
> No, the token replacement operation works for both single-token words and multi-token words. Our token replacement operation masks a word and uses the RoBERTa model to generate corruption. Therefore, the token replacement operation also works for multi-token words.
>
> Question C:
> Yes, we will include it in the revision. The reason for squaring the probability is that it has been normalized by the frequency per class f_c as shown in Equation 2. The intuition for Equation 2 is described in Xie et al. (2016)’s work in Section 3.1.2. The soft-frequency formula is designed to:
> 1. Strengthen predictions.
> 2. Place more emphasis on data points with high confidence.
> 3. Normalize the loss contribution of each centroid to prevent large clusters from distorting the hidden feature space.
>
> Missing Reference:
> Thank you very much, we will update Table 1 and its description paragraph in Section 4.2 accordingly.
>
> References:
>
> Junyuan Xie, Ross Girshick, and Ali Farhadi. Unsupervised deep embedding for clustering analysis. In ICML 2016.
>
> Sascha Rothe, Jonathan Mallinson, Eric Malmi, Sebastian Krause, and Aliaksei Severyn. 2021. A simple recipe for multilingual grammatical error correction. In ACL 2021.
>
> Alexey Sorokin. Improved grammatical error correction by ranking elementary edits. In EMNLP 2022.

---

### Meta-Review · Area_Chair_ZhYi · 2023-09-20

**Recommendation:** 4

**Metareview:**

The paper proposes an unsupervised approach to Grammatical Error Correction (GEC) based on method proposed in Yasunaga et al., (2021). The approach relies on using a large pre-trained language model (Flan T5-xxl) and proposes several novel techniques for synthetic data generation and enhancements to the critic component. Experiments on English and Chinese benchmarks demonstrate superior performance compared to unsupervised baselines and the LM-Critic pipeline.

The paper is clear, well-written, and demonstrates strong results on two languages. Two concerns raised in the reviews are the limited novelty (the paper is an extension of the Yasunaga paper), and the use of large backbone models. It is not clear whether the method can be effective when used with low-resource languages and smaller language models.

---

### Decision · Program_Chairs · 2023-10-07

**Decision:**

Accept-Main

**Comment:**

The paper proposes an unsupervised approach to Grammatical Error Correction (GEC) based on method proposed in Yasunaga et al., (2021). The approach relies on using a large pre-trained language model (Flan T5-xxl) and proposes several novel techniques for synthetic data generation and enhancements to the critic component. Experiments on English and Chinese benchmarks demonstrate superior performance compared to unsupervised baselines and the LM-Critic pipeline.

The paper is clear, well-written, and demonstrates strong results on two languages. Two concerns raised in the reviews are the limited novelty (the paper is an extension of the Yasunaga paper), and the use of large backbone models. It is not clear whether the method can be effective when used with low-resource languages and smaller language models.